# Prevalence of Everyday Discrimination and Relation with Wellbeing among Aboriginal and Torres Strait Islander Adults in Australia

**DOI:** 10.3390/ijerph18126577

**Published:** 2021-06-18

**Authors:** Katherine A. Thurber, Emily Colonna, Roxanne Jones, Gilbert C. Gee, Naomi Priest, Rubijayne Cohen, David R. Williams, Joanne Thandrayen, Tom Calma, Raymond Lovett

**Affiliations:** 1National Centre for Epidemiology and Population Health, Australian National University, Canberra, ACT 2600, Australia; Emily.Colonna@anu.edu.au (E.C.); Roxanne.Jones@anu.edu.au (R.J.); Rubi.Cohen@anu.edu.au (R.C.); Joanne.Thandrayen@anu.edu.au (J.T.); Raymond.Lovett@anu.edu.au (R.L.); 2Department of Community Health Sciences, Fielding School of Public Health, University of California, Los Angeles, CA 90024, USA; gilgee@ucla.edu; 3Centre for Social Research and Methods, College of Arts and Social Sciences, Australian National University, Canberra, ACT 2600, Australia; naomi.priest@anu.edu.au; 4Population Health, Murdoch Children’s Research Institute, Royal Children’s Hospital, Parkville, VIC 3052, Australia; 5Department of Social and Behavioral Sciences, Harvard T. H. Chan School of Public Health, Boston, MA 02115, USA; dwilliam@hsph.harvard.edu; 6Poche Indigenous Health Network New South Wales, University of Sydney, Camperdown, NSW 2006, Australia; tom.calma@health.gov.au; 7University of Canberra, Bruce, ACT 2617, Australia; 8Ninti One, Hackney, SA 5071, Australia

**Keywords:** racism, indigenous peoples, health inequalities, social epidemiology, social determinants of health, Australia

## Abstract

Discrimination is a fundamental determinant of health and health inequities. However, despite the high prevalence of discrimination exposure, there is limited evidence specific to Indigenous populations on the link between discrimination and health. This study employs a validated measure to quantify experiences of everyday discrimination in a national sample of Aboriginal and Torres Strait Islander (Australia’s Indigenous peoples) adults surveyed from 2018 to 2020 (≥16 years, *n* = 8108). It quantifies Prevalence Ratios (PRs) and 95% Confidence Intervals (CIs) for wellbeing outcomes by level of discrimination exposure, and tests if associations vary by attribution of discrimination to Indigeneity. Of the participants, 41.5% reported no discrimination, 47.5% low, and 11.0% moderate-high. Discrimination was more commonly reported by younger versus older participants, females versus males, and those living in remote versus urban or regional areas. Discrimination was significantly associated in a dose-response manner, with measures of social and emotional wellbeing, culture and identity, health behaviour, and health outcomes. The strength of the association varied across outcomes, from a 10–20% increased prevalence for some outcomes (e.g., disconnection from culture (PR = 1.08; 95% CI: 1.03, 1.14), and high blood pressure (1.20; 1.09, 1.32)), to a five-fold prevalence of alcohol dependence (4.96; 3.64, 6.76), for those with moderate-high versus no discrimination exposure. The association was of consistent strength and direction whether attributed to Indigeneity or not—with three exceptions. Discrimination is associated with a broad range of poor wellbeing outcomes in this large-scale, national, diverse cohort of Aboriginal and Torres Strait Islander adults. These findings support the vast potential to improve Aboriginal and Torres Strait Islander peoples’ wellbeing, and to reduce Indigenous-non-Indigenous inequities, by reducing exposure to discrimination.

## 1. Introduction

Racism is a fundamental determinant of health, contributing to health inequities globally [1]. Racism is an organised social system that operates on multiple levels to reinforce, justify, and perpetuate a racial or ethnic hierarchy that devalues, disempowers, and differentially allocates resources to groups defined as “inferior” or “superior” [2,3,4,5]. Racist beliefs and attitudes can be expressed through stereotyping and prejudice, and can manifest as discrimination (unjust treatment) at the intrapersonal, interpersonal, and institutional levels [6]. Discrimination can occur across different life stages, as single instances to repeated exposure, with cumulative effects [4]. Aboriginal and Torres Strait Islander peoples (Indigenous Australians) were constructed as “inferior” through Australia’s colonial era to justify dispossession of their land by settler colonial governments. This racial hierarchy was entrenched through ongoing discriminatory policies and the media which have maintained a system of oppression [7,8,9]. While acknowledging the varied levels, systems, and experiences of racism, and their likely impacts on ill health [3,10], this paper focuses on the individual’s perceived experiences of interpersonal discrimination.

Discrimination is known to be negatively associated with wellbeing, regardless of the perceived reason for the discrimination (i.e., on the basis of race or other characteristics) [4,10,11,12,13]. Interpersonal discrimination can activate stress and negative emotions, which can contribute to mental and physical ill health [1,2,3,5]. Individual responses to discrimination can include health risk behaviours, which in turn decrease health [5]. Discrimination can also lead to reduced access to resources that optimise health, and increased exposure to risk factors for ill health (for example, by contributing to inequities in social determinants of health such as employment) [1,2,3,14]. 

Studies internationally point to a wide range of negative health-related outcomes associated with experiences of discrimination [5]. These include health risk behaviours (low health seeking, low adherence to medical advice, high-risk behaviours, tobacco use, and alcohol and other drug use), poor general mental health and wellbeing (e.g., psychological distress, low life satisfaction, low self-esteem, and stress), mental health conditions (e.g., anxiety, depression, post-traumatic stress disorder, and psychosis), poor physical health, preclinical outcomes, and health conditions (e.g., high blood pressure, heart conditions, diabetes, high cholesterol, cancer, overweight, and obesity) [5]. Studies internationally have also demonstrated associations between discrimination and cultural outcomes such as an individual’s relationship to their racial or ethnic identity [15]. 

Studies among Aboriginal and Torres Strait Islander adults have found associations between experiences of racial discrimination and outcomes related to social and emotional wellbeing (SEWB; general mental health [6,16,17,18,19], anxiety [17], depression [17,20], suicide risk [17], psychological distress [21]), health behaviours (smoking and alcohol use [22]), and physical health (self-reported general health [18] and physical health [19]). These studies have been restricted to experiences of discrimination that are attributed to Indigeneity. We are unaware of any studies with Aboriginal and Torres Strait Islander adults that examine specific physical health conditions or cultural wellbeing outcomes (e.g., connection to culture, identity, and self-determination) [15]. Many Aboriginal and Torres Strait Islander peoples hold holistic views of wellbeing, and culture is a key determinant of wellbeing [23]. It is therefore critically important that frameworks for considering discrimination and its impacts are inclusive of culture.

Aboriginal and Torres Strait Islander peoples have reported experiencing various types of discrimination [24]. The 2014–2015 National Aboriginal and Torres Strait Islander Social Survey (NATSISS) found that 33.5% of Aboriginal and Torres Strait Islander people aged ≥ 15 years had experienced unfair treatment in the previous year because they were Aboriginal and/or Torres Strait Islander (racial discrimination). Racial discrimination was commonly reported by females and males (35.3% and 31.7%, respectively) [25], across ages (36% of people aged 15–29, 40% aged 30–44, and 31% aged ≥ 45 years) [26], and non-remote (34.9%) and remote (28.2%) areas [25]. In localised studies of Aboriginal and Torres Strait Islander adults, the prevalence of experiences of discrimination varies from a third [17] to almost all (97%) [21] participants [6,16,19,27]. Variation may be influenced by study location, sample size and selection, and the discrimination measures used.

Discrimination is a common experience for Aboriginal and Torres Strait Islander peoples and is associated with negative SEWB and physical health outcomes. Accordingly, Aboriginal and Torres Strait Islander stakeholders have identified that understanding experiences of discrimination is of high research priority [2,28,29]. However, to date, the majority of studies have been localised with small sample sizes (approximately 100–350 participants), have not used measures validated for use in diverse Aboriginal and Torres Strait Islander contexts, often rely on single item or brief measures, and have examined a limited number of outcomes. Further, we lack evidence on the totality of discrimination experiences in the population, and whether the impacts of discrimination vary by attribution to Indigeneity. Using data from Mayi Kuwayu: The National Study of Aboriginal and Torres Strait Islander Wellbeing (the Mayi Kuwayu Study) [30], this paper aims to: quantify experiences of everyday discrimination in a national sample of Aboriginal and Torres Strait Islander adults (≥16 years), overall and by key characteristics; apply an outcome-wide approach [31] to quantify the relationship between experiences of discrimination and wellbeing outcomes; and test whether associations between experiences of discrimination and wellbeing outcomes vary by attribution of discrimination to Indigeneity.

## 2. Materials and Methods

### 2.1. Study Population

The Mayi Kuwayu Study is a national longitudinal study of Aboriginal and Torres Strait Islander adults aged ≥ 16 years living in diverse settings across Australia. Baseline data collection commenced in 2018, and recruitment is ongoing. The current cross-sectional analysis is based on Mayi Kuwayu Study Release 2.0, which includes survey responses scanned and received by 1 May 2020 (*n* = 9691). The study uses multi-mode recruitment; this data release includes participants from all modes of survey completion: paper questionnaire, online, and face-to-face with the support of a Mayi Kuwayu Study team member. Study details are provided elsewhere [30,32].

### 2.2. Data

All data analysed in the current study are based on self-reported responses to the baseline questionnaire. Detailed variable definitions are provided in Appendix A.

### 2.3. Everyday Discrimination

The eight-item everyday discrimination instrument used in the Mayi Kuwayu Study was based on existing measures (including the Everyday Discrimination Scale), adapted to diverse Aboriginal and Torres Strait Islander contexts through an iterative community consultation process, field testing, and stakeholder input [30,33]. Participants are asked to report the extent to which they experience discrimination in eight specific settings (see Appendix A), with the response options of: “not at all” (coded as 0), “a little bit” (1), “a fair bit” (2), “a lot” (3). A total score is formed for those with complete data, summing responses, and categorised as no (score 0/24), low (1–8/24), or moderate to high (9–24/24) discrimination. 

Participants then answer a global attribution question: “When these things happen, do you think it is because you are Aboriginal/Torres Strait Islander?”, with the same response options. Participants were defined as attributing experiences to Indigeneity if they responded “a little bit” to “a lot”. Participants who did not answer the question are coded as missing data on attribution.

To enable comparison of discrimination experiences attributed versus not attributed to Indigeneity, we created a composite variable categorized as: no discrimination, low discrimination without attribution, low discrimination with attribution, moderate/high discrimination without attribution, moderate/high discrimination with attribution, or missing. Participants with missing data on discrimination total score and/or on attribution are coded as missing.

### 2.4. Demographic Factors

Data are presented by age group (16–35, 36–55, ≥56), gender (male, female, or another gender; participants identifying as another gender are included in all analyses except those stratified or adjusted for gender, to protect confidentiality and avoid small cells), remoteness (major cities, inner and outer regional, remote and very remote—coded according to the ASGS Remoteness Structure [34]), State/Territory of residence, highest educational qualification (less than Year 10, Year 10, Year 12, or beyond), and family financial situation (“run out of money or spend more than we get”, “just enough money”, “some or a lot of savings”).

### 2.5. Wellbeing Outcomes

We examined the relationship between everyday discrimination and a range of wellbeing indicators identified in the literature as associated, or considered to be conceptually related, to discrimination. This included measures of SEWB (pain, life satisfaction, happiness, psychological distress, doctor-diagnosed anxiety, depression), culture and identity (control over life, feeling torn between Aboriginal and/or Torres Strait Islander and non-Indigenous culture, feeling disconnected from Aboriginal and/or Torres Strait Islander culture, choosing not to identify as Aboriginal and/or Torres Strait Islander in the Census, study, work, Centrelink, or real estate), health behaviours (smoking status, gambling, alcohol dependence), and physical health outcomes (general health, doctor-diagnosed heart disease, high blood pressure, high cholesterol, diabetes). Each outcome is coded as a binary variable for analysis, collapsed from ordinal item response options for most variables (more detail in Appendix A). 

### 2.6. Sample

Participants were excluded from the current analysis if they were missing data on: total discrimination score (*n* = 765/9691), age (*n* = 667/8926), gender (*n* = 51/8259), or remoteness (*n* = 100/8208), leaving 83.7% of the original sample (*n* = 8108/9961).

### 2.7. Statistical Analysis

The distribution of the sample across demographic factors is presented. Total discrimination scores (mean and 95% Confidence Interval, CI, and distribution across categories) and scores for individual items (mean, 95% CI and percentage reporting any experience) are presented overall and in relation to demographic factors. For those experiencing low and moderate-high levels of everyday discrimination, the extent of attribution to Indigeneity is presented.

We use binomial regression to calculate Prevalence Ratios (PRs) and 95% Confidence Intervals (CIs) for each outcome in relation to total discrimination score category. The significance of overall coefficients is tested using the Wald test. To test for a dose-response relationship (trend), models are re-run with the categorical discrimination variable included as a continuous variable, re-coded as the mean total score for each discrimination category. Models are restricted to participants with data on the outcome of interest. Models are presented unadjusted, and then adjusted for age group, gender, remoteness, as these factors were identified a priori as potential confounders of the relationship between discrimination experiences and wellbeing outcomes; adjusted models are reported in the text. Given the potentially clustered geographic distribution of participants, models were re-run accounting for clustering using Indigenous Region (IREG) [34] as the cluster variable. Results were not materially different after accounting for geographic clustering (data not shown).

To understand the role of attribution in the links between discrimination experiences and wellbeing outcomes, models were re-run using the composite measure of total discrimination score and attribution as the exposure variable. Given the potential for confounding by socioeconomic status, we repeated all models additionally adjusted for education and additionally adjusted for financial status. Cells <5 are confidentialised, with the exception of cells for the missing category (which do not pose any risk of identification). Across analyses, an alpha level of 0.05 was the threshold for statistical significance.

### 2.8. Ethics

Participation in the Mayi Kuwayu Study was voluntary and with informed consent. The Mayi Kuwayu Study is Aboriginal-led and governed, and conducted with ethics approvals from national, State and Territory Human Research Ethics Committees (HRECs) and from relevant Aboriginal and Torres Strait Islander organizations. This study was conducted under The Australian National University HREC protocol 2016/767, and with approval from the Mayi Kuwayu Study Data Governance Committee (Project D200506).

Early results were discussed with the Thiitu Tharrmay Research Reference Group (August, December 2020) and with the Australian Institute of Aboriginal and Torres Strait Islander Studies (November 2020) to inform interpretations and future directions. Findings from these stakeholder discussions are incorporated throughout the discussion to contextualise findings. Aboriginal and/or Torres Strait Islander peoples were involved at all research stages. 

## 3. Results

Of the sample, 61.6% were female and 38.3% male; 42.9% were from major cities, 47.8% regional, and 9.3% remote; 25.7% were aged 16–35 years, 34.9% 36–55, and 39.4% ≥ 56 (Table 1). The highest percentage of participants were from NSW (34.3%) and Queensland (26.7%), with ≤11% of the sample in each of the other States/Territories.

The mean total discrimination score was 3.19 (95% CI: 3.10, 3.29), with 41.5% reporting no discrimination, 47.5% low, and 11.0% moderate-high discrimination (Table 1). The mean score was significantly lower for the ≥56-years age group compared to younger age groups, females compared to males, for major city and regional compared to remote participants, and for those with the highest levels compared to lower levels of education and financial status. The mean score was lowest among participants living in Tasmania (1.75; 1.43, 2.07) and highest in NT (4.56; 4.11, 5.01) and WA (4.99; 4.64, 5.35).

In the whole sample, the mean was highest for the items “People act like I am not smart” (0.63; 0.61, 0.64) and “I am treated with less respect than other people” (0.57; 0.55, 0.59), with 42.2% and 42.0% respectively endorsing these items “a little bit” to “a lot” (Table 2). Means for the other items ranged from 0.25 to 0.41. The mean score for all individual items was significantly lower for the oldest (≥56 years) compared to younger age groups. The mean for individual items was generally similar for males and females, with the exceptions of “People act like they think that I am not smart” which was higher among females, and “People act like they are afraid of me” and “Police unfairly bother me” which were higher among males. Many items (treated with less respect, worse service, called names, police) had a higher mean score among participants living in remote compared to major cities or regional areas.

Attribution to Indigeneity was more common among those experiencing moderate-high compared to low levels of discrimination, with 90.6% versus 63.5% reporting any attribution. 

We observed strong evidence of associations between everyday discrimination and each outcome (representing poor wellbeing) examined (Figure 1; Appendix A). There was a significant trend for all outcomes, with increasing discrimination exposure associated with increasing outcome prevalence. For SEWB outcomes, the PR for those experiencing moderate/high, compared to no discrimination was 3.74 (95% CI: 3.11, 4.49) for low happiness, 3.44 (3.10, 3.82) for low life satisfaction, 2.48 (2.29, 2.69) for high/very high psychological distress, 1.64 (1.53, 1.77) for frequent experience of pain, 1.63 (1.49, 1.79) for depression, and 1.60 (1.44, 1.78) for anxiety. Corresponding PRs for culture and identity outcomes were 3.77 (3.27, 4.34) for low control over life, 1.77 (1.47, 2.14) for choosing not to self-identify as Aboriginal and/or Torres Strait Islander, 1.69 (1.58, 1.82) for feeling torn between cultures, and 1.08 (1.03, 1.14) for feeling disconnected from Aboriginal and/or Torres Strait Islander culture. For health behaviour outcomes, the respective PRs were 4.96 (3.64, 6.76) for ever having alcohol dependence, 2.21 (1.99, 2.46) for being a current smoker, and 1.10 (1.02, 1.20) for gambling in the last year. Finally, for physical health outcomes, the corresponding PRs were 2.16 (1.97, 2.37) for poor/fair general health, 1.52 (1.30, 1.78) for diabetes, 1.50 (1.22, 1.85) for heart disease, 1.20 (1.09, 1.32) for high blood pressure, and 1.15 (1.02, 1.29) for high cholesterol.

The overall pattern of association was similar when outcomes were examined in relation to the composite measure of discrimination experience and attribution (Figure 2; Appendix A); while all variables were still significant, some individual coefficients were no longer significant, and the smaller numbers in exposure categories (particularly moderate-high discrimination without attribution) led to wider confidence intervals. In general, the strength of association was similar regardless of the attribution of Indigeneity; that is, confidence intervals surrounding coefficients overlapped for those experiencing low discrimination versus without attribution, and for those experiencing moderate/high discrimination versus without attribution. There were three exceptions, where the effect varied by attribution: the relationship of pain to low discrimination without attribution (PR = 1.45; 1.35, 1.55) was stronger than for low discrimination with attribution (PR = 1.20; 1.13, 1.28), the relationship of feeling torn between cultures to low discrimination with attribution (PR = 1.69; 1.60, 1.79) was stronger than for low discrimination without attribution (PR = 1.45; 1.35, 1.55), and the relationship of gambling to high discrimination with attribution (PR = 1.14; 1.05, 1.24) was in the opposite direction to that for high discrimination without attribution, where the PR was significantly less than 1 (PR = 0.71; 0.50, <1.00).

There remained strong evidence of association between discrimination and wellbeing outcomes (with the exception of cholesterol) after adjustment for financial status, but many associations were attenuated; for example, the PR for alcohol dependence changed from 4.96 (95% CI: 3.64, 6.76) to 3.47 (2.50, 4.80) (Appendix A). No material change was observed after adjustment for education (Appendix A). A similar pattern of findings with additional adjustment was observed for discrimination with/without attribution (Appendix A).

## 4. Discussion

These findings provide quantitative evidence to underpin the widely held notion that discrimination is commonly experienced by Aboriginal and Torres Strait Islander peoples and is associated with a broad range of negative wellbeing outcomes. These population-specific data can empower communities by informing local action. These findings strengthen the evidence base through the use of a measure developed and validated with and for Aboriginal and Torres Strait Islander peoples [35]; inclusion of a diverse and national sample [32]; and holistic and outcome-wide approach, including outcomes meaningful to Aboriginal and Torres Strait Islander peoples. These outcomes span SEWB, culture and identity, health behaviours, and health outcomes, and include subjective measures of wellbeing and clinical outcomes. Across the outcomes examined, attribution of the experiences of discrimination to Indigeneity did not materially change the strength or direction of the association between discrimination experience and the outcome. Hence, discrimination is associated with poor wellbeing in this sample, and this is not limited to experiences that participants attribute to their Indigeneity. While these results do not provide evidence of a causal relationship, the high prevalence of discrimination and strong and wide-ranging links to wellbeing supports the vast potential to improve Aboriginal and Torres Strait Islander peoples’ wellbeing by reducing exposure to discrimination, or, failing that, by mitigating the negative impacts of discrimination where it occurs.

The majority of participants in this sample (58.5%) reported experiencing at least one type of discrimination. While not representative of the total Aboriginal and Torres Strait Islander population or directly comparable to previous estimates (given differences in the measures used and sampling design), these findings demonstrate high exposure to discrimination within this diverse national sample. The types of discriminatory experiences most commonly reported were people acting as if the participant was not smart or treating them with less respect than other people, with 4 in 10 participants experiencing these at least “a little bit”. Being unfairly bothered by the police was the least commonly reported experience, but still reported by 1 in 6 respondents; this is far from inconsequential given that unfair treatment by police is an extreme form of interpersonal discrimination, with potential substantial implications for life opportunities. 

Within this sample, discrimination was more commonly reported by younger compared to older participants, consistent with findings from the 2014–2015 NATSISS on unfair treatment due to Indigeneity in the past year [26]. Given that our study captured lifetime exposure to discrimination, we would have expected to see increasing prevalence with age, in contrast to the observed findings. Stakeholders interpreted this result to reflect a normalising bias (i.e., “you get used it to”) or conflicting concerns for older people (i.e., “more important things to worry about”). These stakeholder views are supported by previous qualitative research wherein older participants described ignoring or minimising experiences of racism in later stages of life [36], or experiencing such pervasive racism “since they were born” that it was more difficult to perceive discrimination (p. 4, [24]).

Within the Mayi Kuwayu Study sample, discrimination was more commonly reported by those living in remote settings than in major cities or regional areas. This contrasts with findings from the 2014–2015 NATSISS, which found a higher overall prevalence in non-remote versus remote settings of unfair treatment due to Indigeneity in the past year [25,37]. The observed discrepancy could reflect differences in lifetime versus recent exposure. The differences could also potentially reflect differences by remoteness in propensity to attribute experiences of discrimination to Indigeneity, as the NATSISS items were restricted to discrimination experiences attributed to Indigeneity. In the face validity processes underpinning development of the everyday discrimination items used in the Mayi Kuwayu Study, underreporting of experiences of discrimination was observed in a remote setting when the survey items explicitly asked about unfair treatment due to Indigeneity [35]. Further, the prevalence estimates from the Mayi Kuwayu Study are not adjusted for potential demographic or other confounders, which may contribute to observed differences in prevalence by remoteness. In the Mayi Kuwayu Study, reporting was particularly elevated in remote compared to other settings for being unfairly bothered by the police. This is consistent with the 2014–2015 NATSISS, wherein the prevalence of being unfairly arrested or charged was significantly higher among participants living in remote compared to non-remote settings [25]. 

The current study provides the most comprehensive exploration to date of a range of wellbeing outcomes in relation to discrimination within the Aboriginal and Torres Strait Islander population. It supports previous population-specific evidence on a relationship between discrimination and SEWB (general mental health [6,16,17,19], anxiety [17], depression [17,20], psychological distress [21]), physical health (general health [19]), and health behaviours (smoking and alcohol use [22]), and provides the first population-specific evidence on the link between discrimination and key health conditions (heart disease, high cholesterol, high blood pressure, diabetes), health behaviours (gambling), and culture and identity outcomes (low control over life, feeling torn between cultures, feeling disconnected from culture, and choosing not to self-identify as Aboriginal and/or Torres Strait Islander) for adults. 

The strength of the association between discrimination and wellbeing outcomes varied across outcomes, consistent with international evidence [14,15,38]. The strongest link was found for the outcome of alcohol dependence, where prevalence was five-fold among those experiencing moderate-high compared to no discrimination. A previous study in Victoria found that the unadjusted odds of experiencing racism were increased for participants abstaining from alcohol compared to consuming alcohol in the past year (OR = 1.3; 95% CI: 1.2, 1.5) and consuming ≥4, compared to 1–2, standard drinks in a typical drinking session (OR = 1.4; 95% CI: 1.3, 1.6); we are unaware of any other evidence specific to the Aboriginal and Torres Strait Islander population for comparison. There is currently no validated measure of alcohol dependence for the Aboriginal and Torres Strait Islander population [39], and there may be limitations to the measure used in this study (see Appendix A). An international systematic review provides some evidence of a positive association between discrimination and alcohol consumption, drinking-related problems, and risk of disorders; however, findings were inconsistent, varying by study type and outcome examined [40]. This association should be explored in further detail, given the strength of association observed and high burden of alcohol-related morbidity and mortality within the Aboriginal and Torres Strait Islander population [41]. 

We also observed a strong association with current smoking status, with smoking over twice as common among those experiencing moderate-high levels of discrimination, compared to no discrimination (PR = 2.21;1.99,2.46). The strength of the association between discrimination and key health risk behaviours [41]—smoking and alcohol dependence—indicates the importance of considering discrimination as a risk factor in prevention and treatment [15]; these factors have previously been conceptualised as risk factors for experiencing discrimination [22].

Discrimination experiences were strongly associated with SEWB outcomes in this study. It was three to four times as common for those experiencing moderate-high, compared to no discrimination to report low life satisfaction and low happiness. Prevalence was around two-fold for other SEWB-related outcomes: psychological distress (according to the Kessler-5), pain, and doctor-diagnosed anxiety and depression. Noting the lack of direct comparability of findings due to differences in exposure and outcome definition, the pattern of findings is generally consistent with Aboriginal and Torres Strait Islander specific evidence on links between discrimination and SEWB-related outcomes (including the Kessler-5 [21], SF-12 mental component score [6,16,19], Strong Souls mental health score [17], anxiety [17], depression [17,20], suicide risk [17] and general mental health [18]), and international evidence identifying stronger links between discrimination and mental health-related (compared to other) outcomes [14]. 

Consistent with previous international evidence [14], we found a stronger association between discrimination and general physical health (PR = 2.16;1.97,2.37) than specific physical health outcomes (PR = 1.20 for high blood pressure to 1.52 for diabetes). Studying the effects of discrimination on single outcomes may therefore underestimate the effect of discrimination on health. While the strength of association may appear modest (up to 50% higher prevalence) for high blood pressure, high cholesterol, heart disease, and diabetes, these links are important given the high burden of cardiovascular disease and diabetes in the population [41], and to our knowledge have not been quantified previously. The magnitude of association with general health is consistent with previous research in the population [18,19].

We found that experiences of discrimination were associated with indicators of reduced cultural wellbeing: low control over life, feeling torn between Aboriginal and Torres Strait Islander and non-Indigenous culture, and feeling disconnected from Aboriginal and/or Torres Strait Islander culture. The association was particularly strong for the outcome of low personal control (PR = 3.77; 3.27, 4.34). This outcome is intended to serve as an indicator of self-determination, a key cultural domain [23], noting the absence of a validated measure of personal control for Aboriginal and Torres Strait Islander peoples [42]. Feelings of lack of control over one’s life may be reflective of the historic and ongoing colonial processes that were designed to control Aboriginal and Torres Strait Islander lives [43,44]. Further research with Aboriginal and Torres Strait Islander peoples is needed to understand these experiences. We also found that experiences of discrimination were linked to participants choosing not to self-identify as Aboriginal and/or Torres Strait Islander in the Census, at study, at work, with Centrelink, or with real estate. This may reflect a desire to avoid further experiences of discrimination and may lead to differential under-identification of Aboriginal and Torres Strait Islander peoples in key administrative data collections.

We observed a dose-response relationship across all outcomes examined, with increasing exposure to discrimination associated with increasing outcome prevalence. While this is consistent with a causal relationship, the current analysis is cross-sectional and does not provide evidence of causality. The underlying relationships between experiences of interpersonal discrimination and wellbeing are complex, and further research is required to provide information on directionality and underlying mechanisms. Many of the outcomes examined may be inter-related (e.g., SEWB and physical health outcomes), or may be on a shared pathway (e.g., high cholesterol, high blood pressure, and smoking are risk factors for heart disease). It is possible that the observed associations are in part due to reverse causality, i.e., that Aboriginal and Torres Strait Islander peoples are more likely to experience discrimination and/or to find themselves in settings where discrimination occurs if they exhibit these health behaviours (e.g., smoking, alcohol use), experience poor health and wellbeing, or are disempowered. It is likely that the observed associations include contributions in both directions, and these may be reinforcing. Regardless of the direction(s) of association, the importance of reducing discrimination is clear. 

The association of discrimination with negative wellbeing outcomes was similar for discrimination that was attributed to Indigeneity (i.e., racial discrimination) and discrimination that was not attributed to Indigeneity. This is consistent with the international evidence [12], and, to our knowledge, the first time this has been explored explicitly for any Indigenous population internationally. There were three instances in which the association between discrimination and the outcome varied by attribution (pain, feeling torn between cultures, and gambling). In the case of feeling torn between cultures, it conceptually makes sense that discrimination attributed to Indigeneity might have a stronger impact on the outcome than discrimination not attributed to Indigeneity. Further research is required to understand why a divergent pattern emerged for pain and gambling. The general pattern of findings reinforces the importance of collecting quantitative data on all discrimination experiences, not just those attributed to Indigeneity, as well as qualitative data, to enable comprehensive exploration of experiences and impacts of discrimination for this population. 

In this sample, it was substantially more common for participants to attribute experienced discrimination to Indigeneity when they had higher exposure to discrimination. Attribution is difficult as it is generally not possible to know the reason for the unfair treatment experienced (ambiguity) [11]. Many participants in this sample may experience discrimination on multiple domains (“multiple discrimination” [45]), i.e., in relation to both Indigeneity and gender, disability, sexual orientation, socioeconomic status, or other forms of marginalization. For these participants, it is particularly difficult to identify the source of the attribution. Further, these participants may be discriminated against in relation to a combination of these statuses (“intersectional discrimination” [45]), such as being an Aboriginal female, or being an Aboriginal person who identifies as having a disability. These participants might perceive the discrimination experienced in relation to these multiple axes of inequity, and therefore be less likely to attribute experiences to the single factor of Indigeneity [45]. Future research with this sample can explore experiences and attribution of discrimination to Indigeneity in relation to social categories. In addition, for people with multiple “disadvantaged statuses”, attribution of experiences of discrimination to a single factor can vary by the context in which the discrimination occurred [45]. This is difficult to disentangle in this dataset due the use of a global attribution question; that is, participants were not able to indicate different attribution for different types of discrimination experienced.

## 5. Strengths and Limitations 

A strength of this study is the Aboriginal and Torres Strait Islander leadership of and involvement in all stages of the research. The study includes a large, diverse sample that has been designed with considerable community input. However, the Mayi Kuwayu Study is not designed to be representative of all Aboriginal and Torres Strait Islander adults; as such, the prevalence estimates are not generalisable beyond the study sample. In contrast, internal comparisons—i.e., the findings of association between discrimination and outcomes—are understood to be generalisable beyond the study sample [30].

Due to the exploratory nature of this study and desire to examine a broad range of outcomes, we have conducted multiple comparisons, and findings should be interpreted accordingly [46]. Our models were minimally adjusted and therefore there is potential for confounding; however, interpretation of findings was not altered after additional adjustment for individual socioeconomic measures. 

The everyday discrimination instrument used in this study has demonstrated face validity, acceptability, internal consistency, and construct validity within a subsample of Mayi Kuwayu Study baseline participants [35]. However, we acknowledge that any instrument has limitations in capturing the experiences of interpersonal discrimination. A substantial percentage of respondents (41.5%) in this study did not report experiencing any of the specified types of interpersonal discrimination; this group likely represents a combination of participants who: did not experience these types of interpersonal discrimination, were blunted from recognizing these experiences when they occurred due to frequent exposure, preferred not to disclose experiences, and did not recall experiences. Participants may under-report experiences of interpersonal discrimination for many reasons; in contrast, vigilance bias can lead to increased reporting of discrimination experiences. Regardless, reliance on self-reported experiences of discrimination is appropriate [4,10,12,13]. Further, participants who did not experience any of these forms of interpersonal discrimination are likely to still have experienced systemic or structural forms of racism in their lives. If discrimination prevalence is underestimated in this sample, then we are likely to have underestimated the strength of association between discrimination and wellbeing outcomes.

The Mayi Kuwayu Study discrimination items were designed to capture lifetime discrimination experience, rather than exposure over a specific time period, and were designed to capture subjective assessment of the frequency of occurrence, rather than an objective measure [35]. Based on international evidence [14,38], we may have underestimated the strength of association between discrimination and outcomes by capturing lifetime, rather than recent, exposure. Wellbeing is likely shaped by the interplay of historical and recent experiences of discrimination.

## 6. Conclusions

There is a clear need to reduce experiences of interpersonal discrimination for Aboriginal and Torres Strait Islander peoples. The impacts of interpersonal discrimination need to be considered within the broader system of racism, including the interrelated and reinforcing influences of systemic and structural racism, including their impacts on social determinants of health [5].

## Figures and Tables

**Figure 1 ijerph-18-06577-f001:**
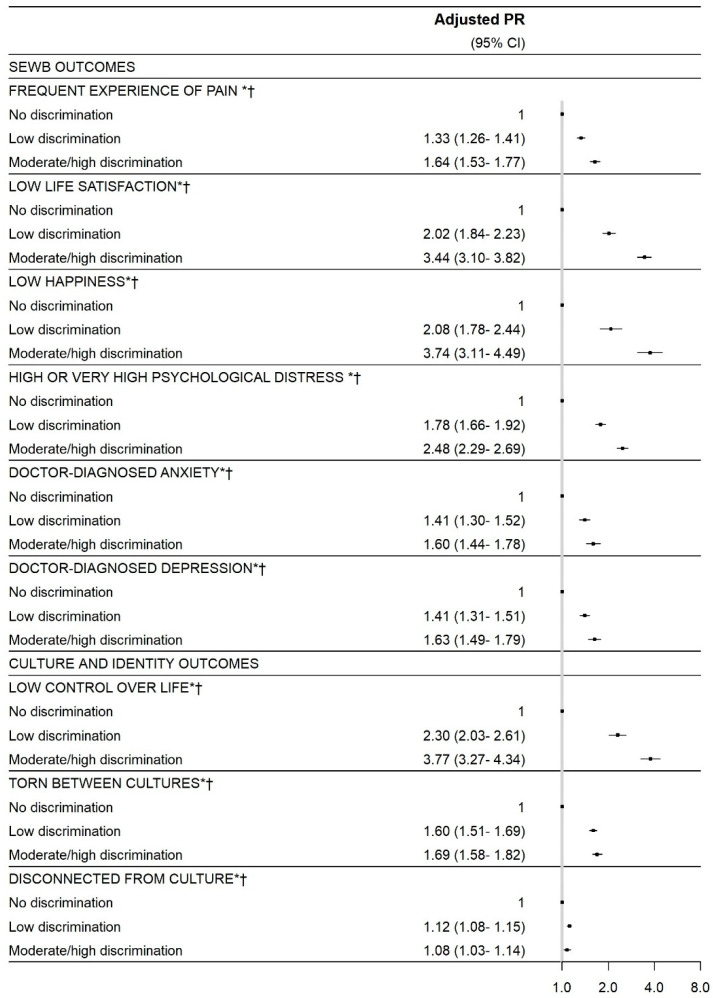
Relationship between experiences of everyday discrimination and wellbeing outcomes (*n* = up to 8100). The adjusted models are adjusted for age group, gender (male or female), and remoteness. Due to small numbers, participants identifying as another gender are excluded from all regression analyses as the fully-adjusted models are adjusted for gender. All models exclude participants missing the outcome of interest.* Indicates that the overall discrimination score variable is significant in the adjusted model, with the *p*-value for the Wald test < 0.05. † Indicates that the trend is significant in the adjusted model (when the discrimination score is included as a continuous variable, set to the mean total score for each discrimination category), with the *p*-value for trend <0.05. **‡** Indicates that the adjusted model for the outcome (heart disease, blood pressure) employs a collapsed age categorisation (16–65 years, ≥66 years) due to small cells. See Appendix A for numbers of participants, unadjusted PR, and crude outcome prevalences by exposure category.

**Figure 2 ijerph-18-06577-f002:**
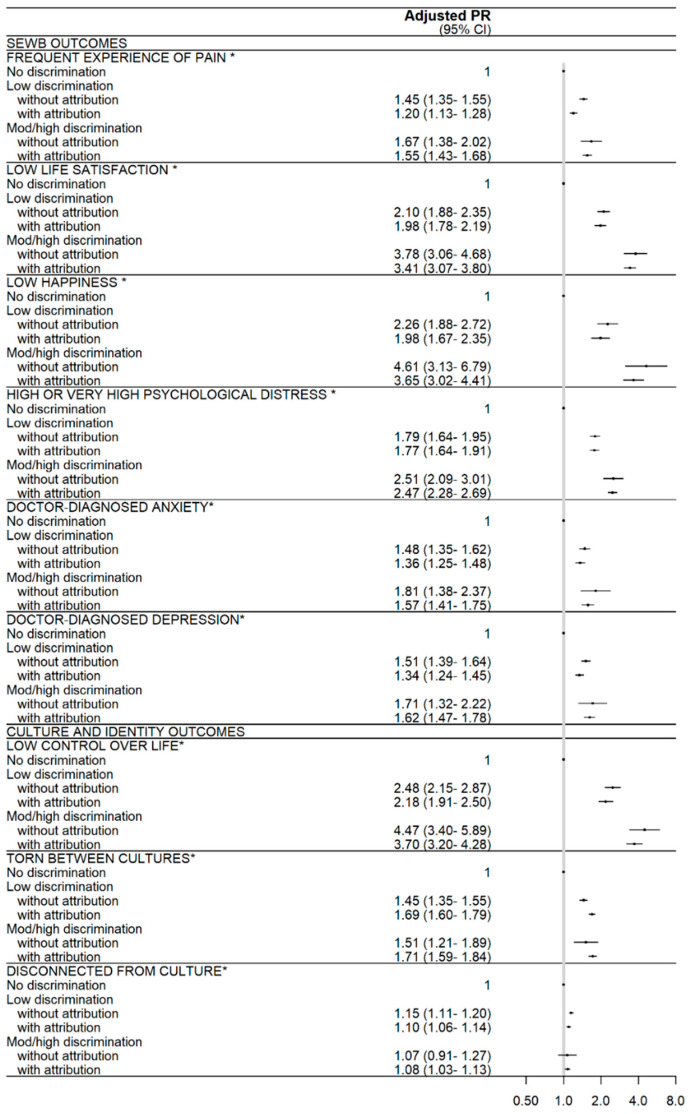
Relationship between experiences of everyday discrimination, with and without attribution to Indigeneity, and wellbeing outcomes (*n* = up to 8039). The adjusted models are adjusted for age group, gender (male or female), and remoteness. Due to small numbers, participants identifying as another gender are excluded from all regression analyses as the models are adjusted for gender. All models exclude participants missing the outcome of interest. * Indicates that the overall discrimination score variable is significant in the adjusted model, with the *p*-value for the Wald test <0.05. † Indicates that the adjusted model for the outcome (heart disease, blood pressure) employs a collapsed age categorisation (16–65 years, ≥66 years) due to small cells. See Appendix A for numbers of participants, unadjusted PR, and crude outcome prevalences by exposure category.

**Table 1 ijerph-18-06577-t001:** Demographic characteristics and experiences of everyday discrimination in the sample (*n* = 8108), and extent of attribution to Indigeneity.

Demographic characteristics	% (*n*)	Everyday Discrimination			
		Mean Score	None	Low	Moderate-High
		(95% CI)	% (*n*)	% (*n*)	% (*n*)
**Overall**	100.0 (8108)	3.19 (3.10, 3.29)	41.5 (3363)	47.5 (3851)	11.0 (894)
**Age group (years)**					
16–35	25.7 (2084)	3.87 (3.67, 4.08)	33.1 (689)	52.8 (1101)	14.1 (294)
36–55	34.9 (2833)	3.99 (3.80, 4.17)	33.1 (937)	52.3 (1481)	14.6 (415)
≥56	39.4 (3191)	2.05 (1.92, 2.17)	54.4 (1737)	39.8 (1269)	5.8 (185)
**Gender**					
Male	38.3 (3104)	3.38 (3.21, 3.55)	42.5 (1320)	44.8 (1390)	12.7 (394)
Female	61.6 (4996)	3.07 (2.95, 3.20)	40.8 (2040)	49.2 (2458)	10.0 (498)
Do not identify as male or female	0.1 (8)	-	-	-	-
**Level of remoteness**					
Major cities	42.9 (3479)	3.20 (3.05, 3.35)	40.4 (1404)	48.4 (1683)	11.3 (392)
Regional	47.8 (3873)	3.03 (2.89, 3.17)	43.5 (1686)	46.4 (1799)	10.0 (388)
Remote	9.3 (756)	3.99 (3.64, 4.34)	36.1 (273)	48.8 (369)	15.1 (114)
**State/Territory**					
ACT	1.5 (124)	3.31 (2.60, 4.02)	35.5 (44)	55.6 (69)	8.9 (11)
NSW	34.3 (2783)	2.96 (2.80, 3.13)	44.9 (1249)	44.7 (1244)	10.4 (290)
NT	6.2 (500)	4.56 (4.11, 5.01)	30.8 (154)	51.2 (256)	18.0 (90)
QLD	26.7 (2164)	2.87 (2.69, 3.04)	42.7 (923)	48.4 (1048)	8.9 (193)
SA	4.6 (369)	3.39 (2.87, 3.91)	41.5 (153)	47.7 (176)	10.8 (40)
TAS	5.3 (433)	1.75 (1.43, 2.07)	55.9 (242)	40.0 (173)	4.2 (18)
VIC	10.2 (827)	2.68 (2.40, 2.96)	43.5 (360)	47.9 (396)	8.6 (71)
WA	11.2 (908)	4.99 (4.64, 5.35)	26.2 (238)	53.9 (489)	19.9 (181)
**Education**					
Year 12 or beyond	56.0 (4540)	3.00 (2.83, 3.08)	40.9 (1858)	49.8 (2262)	9.3 (420)
Year 10	23.5 (1903)	3.52 (3.31, 3.74)	40.9 (778)	45.8 (871)	13.3 (254)
Less than Year 10	19.5 (1582)	3.48 (3.22, 3.73)	43.9 (695)	42.8 (677)	13.3 (210)
Missing	1.0 (83)	3.40 (2.36, 4.44)	38.6 (32)	49.4 (41)	12.0 (10)
**Financial situation**					
Some or a lot of savings	44.3 (3588)	1.94 (1.84, 2.05)	54.1 (1941)	40.9 (1468)	5.0 (179)
Just enough	31.7 (2567)	3.49 (3.32, 3.67)	34.9 (895)	53.4 (1372)	11.7 (300)
Run out of money	15.5 (1255)	5.54 (5.23, 5.85)	21.6 (271)	55.0 (690)	23.4 (294)
Unsure	5.7 (463)	4.28 (3.78, 4.79)	36.7 (170)	46.4 (215)	16.8 (78)
Missing	2.9 (235)	4.32 (3.56, 5.09)	36.6 (86)	45.1 (106)	18.3 (43)
**Attribution of discrimination to Indigeneity**					
Not at all	-	-	-	36.5 (1389)	9.4 (83)
A little bit	-	-	-	41.2 (1567)	20.1 (177)
A fair bit	-	-	-	12.2 (462)	25.0 (221)
A lot	-	-	-	10.1 (383)	45.5 (402)

Due to small numbers, participants identifying as another gender are not presented in gender-stratified results but are included in analysis for all other variables. The total everyday discrimination score ranged from a possible minimum of 0 to a maximum of 24.

**Table 2 ijerph-18-06577-t002:** Mean (95% CI) and per cent (*n*) endorsing each everyday discrimination items, overall and by demographic characteristics (*n* = 8108).

**Mean (95% CI) for Everyday Discrimination Item**									
	*n*	**I Am Treated with Less Respect than Other People**	**I Receive Worse Service than Other People**	**People Act Like I Am Not Smart**	**People Act Like They Are Afraid of Me**	**I Am Called Names, Insulted, or Yelled at**	**I Am Followed Around in Shops**	**I Am Watched More Closely than Others at Work or School**	**Police Unfairly Bother Me**
**Overall**	8108	0.57 (0.55,0.59)	0.40 (0.39,0.42)	0.63 (0.61,0.64)	0.41 (0.39,0.42)	0.31 (0.30,0.33)	0.34 (0.32,0.35)	0.30 (0.28,0.31)	0.25 (0.23,0.26)
**Age group (years)**									
16–35	2084	0.63 (0.60,0.67)	0.44 (0.41,0.48)	0.75 (0.71,0.79)	0.46 (0.42,0.49)	0.43 (0.40,0.46)	0.46 (0.43,0.50)	0.38 (0.34,0.41)	0.32 (0.29,0.35)
36–55	2833	0.70 (0.67,0.73)	0.52 (0.49,0.55)	0.74 (0.71,0.78)	0.53 (0.49,0.56)	0.36 (0.34,0.39)	0.43 (0.40,0.46)	0.38 (0.35,0.41)	0.33 (0.30,0.35)
≥56	3191	0.41 (0.39,0.43)	0.27 (0.25,0.29)	0.44 (0.41,0.46)	0.27 (0.24,0.29)	0.19 (0.17,0.21)	0.17 (0.16,0.19)	0.17 (0.15,0.19)	0.13 (0.11,0.15)
**Gender**									
Male	3104	0.55 (0.53,0.58)	0.41 (0.39,0.44)	0.59 (0.56,0.62)	0.50 (0.47,0.52)	0.32 (0.30,0.35)	0.33 (0.31,0.36)	0.31 (0.29,0.34)	0.35 (0.33,0.38)
Female	4996	0.58 (0.56,0.60)	0.39 (0.37,0.41)	0.64 (0.62,0.67)	0.35 (0.33,0.37)	0.30 (0.29,0.32)	0.34 (0.32,0.36)	0.29 (0.27,0.30)	0.18 (0.16,0.20)
**Remoteness**									
Major cities	3479	0.57 (0.55,0.60)	0.38 (0.36,0.40)	0.64 (0.61,0.67)	0.42 (0.39,0.44)	0.31 (0.29,0.33)	0.35 (0.32,0.37)	0.30 (0.28,0.32)	0.23 (0.21,0.25)
Regional	3873	0.54 (0.51,0.56)	0.38 (0.36,0.40)	0.60 (0.58,0.63)	0.39 (0.37,0.42)	0.30 (0.28,0.32)	0.31 (0.29,0.33)	0.28 (0.26,0.30)	0.23 (0.21,0.25)
Remote	756	0.72 (0.65,0.78)	0.61 (0.54,0.67)	0.66 (0.60,0.73)	0.42 (0.37,0.48)	0.40 (0.35,0.46)	0.43 (0.37,0.49)	0.35 (0.30,0.40)	0.40 (0.34,0.45)
**Percent (*n*) reporting any experience of each everyday discrimination item (“a little bit” to “a lot”)**									
	***n***	**I Am Treated with Less Respect than Other People**	**I Receive Worse Service than Other People**	**People Act Like I Am Not Smart**	**People Act Like They Are Afraid of Me**	**I Am Called Names, Insulted, or Yelled at**	**I Am Followed Around in Shops**	**I Am Watched More Closely than Others at Work or School**	**Police Unfairly Bother Me**
**Overall**	8108	42.0 (3403)	29.1 (2357)	42.2 (3423)	27.8 (2252)	22.9 (1853)	21.9 (1778)	18.9 (1536)	15.5 (1257)
**Age group (years)**									
16–35	2084	48.1 (1002)	33.0 (687)	50.1 (1043)	30.9 (644)	30.8 (642)	29.5 (614)	24.7 (515)	19.4 (404)
36–55	2833	50.7 (1436)	37.0 (1048)	49.3 (1396)	35.2 (998)	26.4 (749)	27.5 (779)	23.7 (672)	20.2 (572)
≥56	3191	30.2 (965)	19.5 (622)	30.8 (984)	19.1 (610)	14.5 (462)	12.1 (385)	10.9 (349)	8.8 (281)
**Gender**									
Male	3104	39.5 (1226)	29.3 (909)	40.3 (1250)	33.2 (1031)	24.0 (745)	21.0 (652)	19.8 (615)	21.7 (673)
Female	4996	43.5 (2173)	28.9 (1446)	43.4 (2168)	24.4 (1219)	22.1 (1104)	22.5 (1125)	18.4 (918)	11.7 (582)
**Remoteness**									
Major cities	3479	42.8 (1489)	27.9 (971)	43.1 (1500)	28.0 (975)	22.9 (797)	22.2 (773)	18.9 (656)	14.5 (503)
Regional	3873	40.3 (1562)	28.0 (1084)	41.3 (1601)	27.5 (1063)	22.1 (854)	20.8 (804)	18.4 (712)	14.8 (573)
Remote	756	46.6 (352)	40.0 (302)	42.6 (322)	28.3 (214)	26.7 (202)	26.6 (201)	22.2 (168)	23.9 (181)

Due to small numbers, participants identifying as another gender are not presented in gender-stratified results but are included in analysis for all other variables. The range for individual items was 0 (“not at all”) to 3 (“a lot”).

## Data Availability

The datasets analysed during the current study are available on application to the Mayi Kuwayu Study Data Governance Committee. This governance body oversees and approves applications for data use, in order to maintain the confidentiality of participants, and ensure that all studies using the Mayi Kuwayu data are protective of Aboriginal and Torres Strait Islander data and cultures. The data application process is detailed here: mkstudy.com.au/overview/ (acessed on 19 May 2021).

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
