# Peer review of "Prevalence of Everyday Discrimination and Relation with Wellbeing among Aboriginal and Torres Strait Islander Adults in Australia"

_ijerph, 2021, doi:10.3390/ijerph18126577_

Round 1

Reviewer 1 Report

Thank you for the opportunity to review this important paper. Overall, this is a very well written paper. My comments are minimal, and aim to clarify the paper, and ensure that all findings are easily understood.

  1. Table 1, mean score, can you clarify the range of the score: I assume that it is 0-5, but this is not specified.
  2. Table 2, all means are below 0. Is the mean reported in Table 2 on the same scale as Table 1? These means seem quite low, comparatively.
  3. Figure 1 is difficult to read. I recommend a reconsideration of the format.
  4. The reference on lines 287-88 is not clear. Is it a new paragraph, or a clarification related to Figure 1?
  5. Line 314. Punctuation missing.

Reviewer 2 Report

This is an important and thoughtfully conducted study. The manuscript is comprehensive and well-presented, and I have no major concerns about the findings or the authors’ interpretation thereof. However, in parts the manuscript is rather verbose. Though the journal specifies no word limit, I suggest that a little further editing of the Introduction and Discussion sections for brevity and reduced repetition might enhance readability.   

Methods: The Methods section is thorough, succinct and lucid.

Some minor issues:

The source of the remoteness classification (presumably ASGS) ought to be stated and referenced in the main text [line 165].

It should be made clear within the main text [see lines 179-180] that the binary outcome variables have (mostly) been collapsed into binary form for analysis from (mostly) ordinal item response options.

Results:

The title of Table 2 [lines 254-255] could be simplified and made clearer: Firstly, the statistical material is redundant; secondly, it’s not quite correctly expressed that respondents were ‘endorsing ... items’ [as also appears in Results text, line 245].
